# Multi-Label Metric Learning with Bidirectional Representation Deep Neural Networks

## Abstract

Multi-Label Learning task simultaneously predicting multiple labels has attracted researchers' interest for its wide application. Metric Learning crucially determines the performance of the k nearest neighbor algorithms, the most popular framework handling the multi-label problem. However, the existing advanced multiple-label metric learning suffers the inferior capacity and application restriction. We propose an extendable and end-to-end deep representation approach for metric learning on multi-label data set that is based on neural networks able to operate on feature data or directly on raw image data. We motivate the choice of our network architecture via a Bidirectional Representation learning where the label dependency is also integrated and deep convolutional networks that handle image data. In multi-label metric learning, instances with the more different labels will be dragged the more far away, but ones with identical labels will concentrate together. Our model scales linearly in the number of instances and trains deep neural networks that encode both input data and output labels, then, obtains a metric space for testing data. In a number of experiments on multi-labels tasks, we demonstrate that our approach is better than related methods based on the systematic metric and its extendability.

## 1 Introduction

We consider the problem of multi-label classification, because it wide application. For example, in document classification, one document can cover different topics; in computer vision, one scene image can contain multiple semantic classes like mountain, beach and sea. We address the problem via metric learning approach, where instances with the more different labels move the further away. After the metric learning is completed, the problem is reduced as easy memory-based learning. Thus, the k nearest neighbor (KNN), one of popular and simple methods, can easily handle the multi-label classification. For a new testing instance $x_{test}$, the predicted labels are:

$$y = labels\ with\ most\ instances\ in\{f(x_i)_{i=1}^K\}, i \in Nei(F(x_{test})) \tag{1}$$

$f(\cdot)$ can be a linear mapping, a non-linear RKHS mapping, or neural network-like differentiable function. The performance of kNN significantly relies on the learning of distance metric that computes latent distance between different instances and classes. Kwok and Tsang (Kwok & Tsang, 2003) show that in a single-label prediction task, via pushing two origin nearby instances of different classes further apart with a large margin in RKHS, performance of kNN can be greatly improved. Multi-labels prediction setting, however, restricts the capacity of existing advanced metric learning techniques like the popular LMNN Weinberger & Saul (2009), as different classes can range from a few to hundred different labels, but the appropriate distance metric could not be provided. The recent work of multi-label metric learning (Liu et al., 2018) gave a systematic studies on advanced methods and proposed a new paradigm with better performance, but still suffers following limitations: 1) inferior capacity: the metric learning is based on linear projection, it doesn't learn flexible representation with non-linearity. 2) application restriction: they only handle the hand-craft feature data, but cannon deal with raw image or audio data, or requires pre-processing techniques that may lost key information of the raw data before learning the representation. Therefore, it is non-trivial to break the limitations of existing work and develop capable and extendable framework for solving multi-output learning tasks.

In this work, during training, we project the feature data or directly raw image data into the metric space using neural networks model and incorporate labels information in a bidirectional way. Incorporating labels will exploit the label dependency during learning the metric representation $f(\cdot)$, instead of treating them as independent ones. Integrating $F(\cdot)$ and $G(\cdot)$ in a bidirectional way will allow the model concentrate the instances with identical label and will enable it to get the metric representation of testing data without labels.

Our contributions are two-fold. First, we introduce a simple and well-behaved neural network framework for multi-output task and show how it can motivate via output representation and deep neural networks. Secondly, we demonstrate how this form of neural network framework can be used for scalable and extendable multi-label classification tasks and propose how it extend to other multi-output tasks. Experiments on a number of data sets demonstrate that our model compares favorably both in multi-labels classification performance (micro-F1, example-based F1, precision and accuracy) and extendability (able to modify to more and complex tasks) against state-of-the-art methods for multi-label learning.

## 2 METRIC SPACE WITH BIDIRECTIONAL REPRESENTATION

In this section, we provide motivation for a specific neural network model that we will use in the rest of this paper. We consider a basic and simple neural networks framework with the following propagation:

$$H = \sigma(W_h X + b_h) \tag{2}$$

$$M_x = W_x H + b_x \tag{3}$$
$$M_y = W_y Y + b_y \tag{4}$$

Here, $X$ and $Y$ are the input features and output labels, respectively. The $W$s and $b$s are the weights and bias of corresponding layers. $\sigma(\cdot)$ denotes an activation function, that incorporate the non-linear propagation, such as the $ReLU(\cdot) = max(0, \cdot)$. $M_x$ and $M_y$ have the same shapes, i.e. $X$ and $Y$ are mapped to the same metric space via non-linear multi-layer and single-layer networks. In the following, we show that this basic framework is motivated via outputs representation, and deep convolutional network for visual recognition on multi-label data set.

### 2.1 OUTPUTS REPRESENTATION FOR METRIC LEARNING

We consider a metric space $\mathcal{M}$ in the network embedding defined as $m_X \in \mathbb{R}^d$ (a vector in metric space corresponding to an instance $X$ in input space) with a mapping $F(X)$ parameterized by the networks, i.e.:

$$m_X = F(X) \tag{5}$$

Another important component of the proposed framework is to model the relationship between metric space and the outputs set $Y \in \mathbb{R}^q$. Since mapping of outputs to metric space are generally unknown in advance, we proposed a new representation, namely Reversed Embedding, In order to leverage the $G(Y)$ to bridge the connection between metric space $\mathcal{M}$ and corresponding data $X$ in input space.

$$m_Y = G(Y) \tag{6}$$

Specifically, our intuition is that if the output labels of any two instances $y_i, y_j$ are the same then two corresponding $x_i, x_j$ in the input space should map very closed to each other in the metric space. To capture this intuition, we minimize the following objective function as an bidirectional representations in the same metric space, given by

$$\min_{F,G} \|m_Y - m_X\|_2^2 \tag{7}$$

In a good metric learning, the model should promote discriminative and predictable representation. To obtain a discriminative representation, the instances of the same outputs will not be confused with ones of different outputs. As for a predictable representation, the above approach provides an opportunity to exploit the dependency between input data and outputs in the metric space.

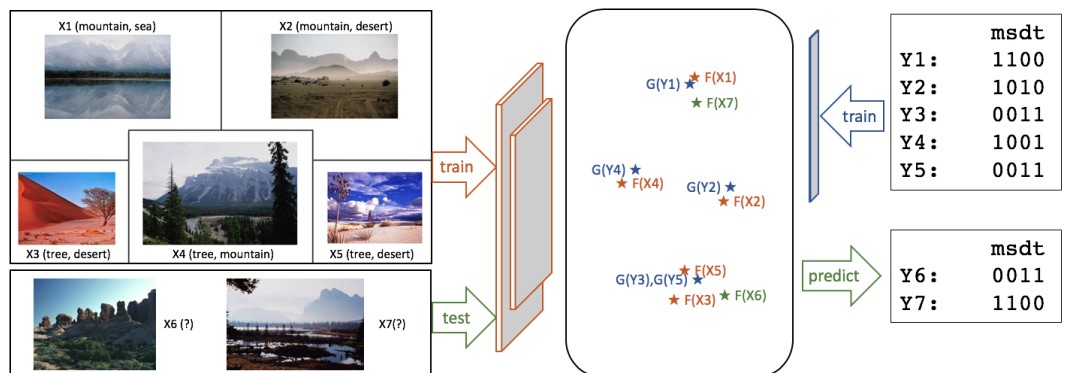

Figure 1: schematic depiction of (Bidirectional Representation Deep Neural Network) BiRDNN for multi-label metric learning. Deep Representation of input $X$ is learned by $F(X)$ into the metric space, where reversed embedding of $Y$ is another representation $G(x)$ from the opposite direction. The letters, m s d t, denote the labels, mountain, see, desert, tree. The corresponding $X$ and $Y$ are mapped to very closed position in the metric space, therefore the different inputs with the same labels will be closed, too. In the testing stage, the predicted label new $X$ is obtained by the labels of the nearest training instances in the metric space.

## 2.2 NEURAL NETWORK FOR DEEP REPRESENTATION

In the last subsection, we mention that the predictable representation requires dependency between input and outputs. However, in real-world application, two instances for the identical outputs may be very distinct in the input space, while the ones with different outputs may sometime look confusing originally, especially for raw image data. This raised issue lights the challenge to learn the $M_X = F(X)$ with sufficient representation capacity. A neural networks model can therefore be built by stacking multiple layers based on the form of Eq 2, where the hidden layer can optionally followed by a point-wise non-linearity. Now, imagine we have to learn an xor mapping from input to metric space, that is non-linear separability, and therefore the non-linear activation like sigmoid or ReLU is applied right after linear mapping to handle it.

Our framework is also extendable, since in this way, we can also learn the unstructured data like image and audio by replacing the linear mapping layer with convolutional layer.

Additionally, for a fixed computational budget, the batch-wise feeding allows us to build a scalable models, a practice that is known to train on a large number of data set. Unlike MMOC and ML-KNN (Zhang & Schneider, 2012; Zhang & Zhou, 2007), there are no quadratic or exponential computational costs during training with respect to the number of instances.

## 3 DEEP METRIC LEARNING FOR MULTI-LABEL CLASSIFICATION

Having introduced a basic, yet extendable framework for efficient information propagation and integration on both input data and outputs, now we can be back to the multi-label classification task. As outlined in the introduction, we project input training data with integration of output label during metric learning, but in a bidirectional way, then get the metric representation of testing data without labels, finally a simple KNN will obtain the predicted labels. The overall model, a Bidirectional Representation Deep Neural Networks (BiRD-NN) for multi-label metric learning, is schematically depicted in Figure 1

### 3.1 SIMPLE EXAMPLE FOR FEATURE INPUT DATA

In the following, we consider $F(X)$ as a two-layer representation network for input $X \in \mathbb{R}^{p \times n}$, i.e. $n$ instances of $p$-feature data. We incorporate the non-linearity $\sigma(\cdot)$ in the hidden layer of $F(x)$, then the forward representation takes this simple form:

$$F(X) = W_x(\sigma(W_h X + b_h) + b_x) \tag{8}$$

Here, $W_h \in \mathbb{R}^{h \times p}$ is an input to hidden weights matrix for a hidden layer of $h$ features. $W_x \in \mathbb{R}^{d \times h}$ is a hidden-to-metric weight matrix and $d$ is the dimension of metric space. $b_h \in \mathbb{R}^h$ and $b_x \in \mathbb{R}^d$ are the corresponding bias vectors. The non-linear activation is defined as $\sigma(\cdot) = ReLU(\cdot) = \max(0, \cdot)$.

For metric learning, we reversely represent the $q$-labels of each instance, and pull the inputs of the same labels together:

$$G(Y) = W_y Y + b_y \tag{9}$$

$$\min_{G,F} \sum_{i=1}^{n} \|F(X_i) - G(Y_i)\|_2^2 \tag{10}$$

where $W_y \in \mathbb{R}^{d \times p}$, $Y \in \mathbb{R}^{p \times m}$ and $b_y \in \mathbb{R}^d$, so with $F(X), G(Y) \in \mathbb{R}^{n \times d}$ and we can calculate the instance-wise L2 distance. As there are instances with the identical labels mapping to identical $G(Y)$, their inputs move closed indirectly via $F(X)$ and the minimizing distance objective. The neural network weights $W$s and bias $b$s are trained using gradient descent. In this work, we can perform batch gradient descent training the full data-set, which is a viable option as long as data sets fit in memory. Using a instance-wise L2 distance, time requirement is $O(n)$, i.e. linear in the number of instance.

Another constraint for metric learning is that the instances with different labels should move far away in the metric space. In the multi-labels setting, the difference between two instances can be a few labels to hundred labels. Therefore, we can define how far away are two instance in the metric space according to the difference in the labels space. Following Liu et al. (2018)'s work, we define it as $\Delta(Y_i, Y_j)$ using the euclidean metric to measure the distance between instances $X_j$ and $X_j$ and then learn a new distance metric space, which improves the performance of discriminative classification. To let different instances have a margin of $\Delta(Y_i, Y_j)$ in the learning metric space, we have the following objective function.

$$\min_{F} \sum_{i}^{n} \max(0, \Delta(Y_i, Y_j) - \|F(X_i) - F(X_j)\|_2^2), j \in Nei(F(X_i)), \forall j \tag{11}$$

Here, $Nei(F(X_i))$ is the k neighbor instance of $F(x_i)$. Note here searching the k nearest instance of $F(x_i)$ has a time computational cost of $O(n^2)$, but with mini-batch stochastic gradient descent training of neural network, we can search neighbors in each mini-batch instead of the whole data set. Let $n_{mini}$ be the number of each mini-batch, then the kNN time cost is $O(n_{mini}^2)$, with $n/n_{mini}$ times of kNN for the whole data set. Therefore the time cost of mini-batch kNN is $O(n_{mini}n)$. If we set a small $n_{mini}$, computational complexity is also linear to the number of instances in training stage (In the testing stage, we utilize ball-tree k-NN Nitin Bhatia (2010)).

We leave efficient-mini-batch and dynamic margin extensions for future work.

## 3.2 Extension For Unstructured Input Data

The aforementioned simple example is a shallow model and the training data are hand-crafted feature instead of original data. Fortunately, our framework can be easily extended to more complex application that is involved with unstructured original data like image. Convolutional Neural Networks (CNN) are designed to conduct a task where training inputs are two-dimensional images for computer vision tasks. CNNs can be directly applied in our framework to implement the $F(X)$. Stacking a bunch of convolution layers to learn representation, it is generally call deep representation learning.

Deep representation learning (Goodfellow et al., 2016) has achieved great success due to it high representation capacity and end-to-end learning manner, i.e. the raw images data are fed to the deep neural networks directly without the hand-crafted feature extraction as a pre-processing. There are batch of image pre-processing methods such as such as SE, SIFT and PCA, converting an image to a feature vector, but they are not about learning and might lose key information of the original image, which is inferior to deep representation method based its richer feature learning capacity.

### 3.2.1 COMPARISON WITH RECENT DEEP MULTI-LABEL LEARNING

In our extended model and the last experimental comparison with a deep multi-label learning model AttCNN (Hand et al., 2018), we implement their convolutional component via 4 stacking convolutional network blocks, each of which contains a 3x3 Convolutional Kernel, BatchNormalization, ReLU and a 2x2 max pool. On the top of the convolutional blocks is 2 layers of full connection networks. This is a tailored and modified version of VGG, but the last layer is changed to make the AttCNN and BiRDNN. 1) the last layer is replaced by a multiple-label(attribution) layer and this model act as a baseline. 2) the last layer is changed to map to a metric space that is shared with the reversed embedding of BiR and the model is call Bidirectional Representation Deep Neural Networks (BiRDNN) for multi-label metric learning. **Limitation** of AttCNN: it learns multiple labels at simultaneously, but it treats them as independent ones, and dose not exploit the label dependency during learn the representation as BiRDNN does. AttCNN also was affected by the global skewed data distribution, while BiRDNN predicts multiple labels by local information in the metric space.

## 4 RELATED WORKS

The fields of conventional multi-output metric learning and recent works on neural network earning deep representation inspire us to propose out model. In what follows, we provide brief overview on related works of both fields.

### 4.1 CONVENTIONAL MULTI-OUTPUT METRIC LEARNING

A number of approaches for multi-output metric learning have been proposed in the past decades, most of which only handle hand-crafted feature data and involve two main approaches: encoding-decoding projection and large margin constraints. Prominent examples for encoding-decoding projection variant include out encoding (Zhang & Schneider, 2012), and canonical correlation analysis (Zhang & Schneider, 2011). Liu & Tsang (2015) and Parameswaran & Weinberger (2010) designed large margin constraints for metric learning on multi-out data.

Recently, attention has shifted to models that can be applied to large-scales data sets. CoHasing (Shen et al., 2018) compressed input and output to compact binary embedding, thus align the input and output space, and the prediction is very efficient. LMMO-kNN (Liu et al., 2018) proposed several strategies to speed up the training and testing time after a systematic review is given. For all these methods, however, a multi-step pipeline is required where each step has to be optimized separately, and most of them only handle hand-crafted feature data.

### 4.2 DEEP METRIC LEARNING OR DEEP MULTI-LABEL LEARNING

Deep networks that not only map from representation to output but also learn representation itself have recently proposed. Learned representation often result in better performance than that can be obtained with hand-designed representation or hand-crafted features. In the context of metric learning on multi-label data set, there are recent researches on the single field of them: deep metric learning or deep multi-label learning.

For deep metric learning, Multi-Manifold Deep Metric Learning (MMDML) (Lu et al., 2015) learn multiple sets of non-linearly map of multi-class images to a shared feature space, under which the margin of different class manifold is maximized. A deep metric learning variant (Niethammer et al., 2019) was proposed to address the image registration in medical image analysis. The sample mining methods for deep metric learning are reviewed and (Online Soft Mining) OSM is proposed in Wang et al. (2019) Lifted Structured Feature Embedding (Oh Song et al., 2016) is proposed to improve the deep metric learning in terms of F1 and NMI measures for clustering task and recall@K score for retrieval tasks.

For deep multi-label learning, SH+CNN (Zhao et al., 2015) is a deep semantic ranking based method image retrieval via learning hash functions on multi-label images, breaking the bottleneck of early ones that only handle the simple binary similarity. LMLE (Huang et al., 2016) and CRL (Dong et al., 2018) are proposed to rectify the class skew distribution when learning on multi-label image data set, but their work treat each label as independent one and process them one by one. Recently, Multi-Attribute CNN (AttCNN) (Hand et al., 2018) in proposed for multi-attribution image classification

where 40 labels of image are trained and test simultaneously in a deep CNN architecture. In this paper, we will use the last one as a baseline in the multi-label image classification task. From the best of our knowledge, we are the first attempt on deep multi-label metric learning that also considers the output dependency during learning the representation.

## 5 EXPERIMENTS

We test our model in a variety of real-world multi-label experiments: We first compare a simple implement of our BiR with several state-of-the-art multi-label metric prediction methods on the a series of conventional feature data. Then we compare our deep representation implement, BiRDNN, with the recent deep multi-label learning on a raw image data set.

### 5.1 DATA SETS

Table 1: Statistics of Datasets for the simple example of BiR

| name | domain | instances | nominal | numeric | labels |
|------|--------|-----------|---------|---------|--------|
| scene | image | 2407 | 0 | 294 | 6 |
| corel5k | images | 5000 | 499 | 0 | 374 |
| delicious | text (web) | 16105 | 500 | 0 | 983 |
| EUR-Lex(dc) | text | 19348 | 0 | 5000 | 412 |
| EUR-Lex(ed) | text | 19348 | 0 | 5000 | 3993 |

For the simple implement example, we closely follow the experimental setup in Liu et al. (2018). Data set statistics are summarized in Tabel 1. All data are available on the website[1].

For the deep deep representation implement BiRDNN, we use the raw image dataset, called scene 2000, that is available online[2] with both processed and raw data set. The data set consists of 2,000 natural scene images each of which that contain one or more of the following objects: desert, mountains, sea, sunset, and trees. Several typical images of this data set has been showed in the Figure 1

### 5.2 EXPERIMENTAL SETUPS

For a simple example, we train a shallow BiR as described in Section 3.1 and systematically evaluate performance by the micro-F1 measure and example-based-F1 measures. These two measures are described and analyzed in Mao et al. (2012) and applied on the same data set and baselines in Liu et al. (2018). 10-fold cross validation is conducted to get the mean and standard deviation.

For the BiRDNN, to make a fair comparison, we implement an identical CNN architecture with that of our baseline AttCNN as the bottom part BiRDNN's $F(X)$. We train both models for a maximum of 200 epochs (training iterations) using Adam (Kingma & Ba, 2014) with a learning rate of 0.001. We initialize weights using the xavier uniform distribution for Full-Connection layers and the normal distribution for Convolution layers. We prevent the models from over-fitting with a Dropout rate of 0.5 in the Full-Connection layers.

### 5.3 BASELINES

We compare our simple implement of BiR against the same baseline methods and the proposed one in Liu et al. (2018), i.e. PLST (Tai & Lin, 2012), MMOC (Zhang & Schneider, 2012), kNN,ML-kNN (Zhang & Zhou, 2007), and LMMO-kNN (Liu et al., 2018).

We further compare our deep representation implement BiRDNN against a recent deep multi-label neural networks, AttCNN, proposed in Hand et al. (2018) over a multi-label image data set. To make a fair comparison, we tailed the AttCNN for these experiments, and implemented AttCNN and BiRDNN with the same convolutional component: 4 convolutional network blocks is stacking,

---

[1]http://mulan.sourceforge.net
[2]http://lamda.nju.edu.cn/files/miml-image-data.rar

and each of them contain a 3x3 convolutional filter, BatchNormalization, ReLU and a 2x2 max pool. The input image is $(128, 128, 3)$ of (width,height,channel), and the numbers of output channels in the convolutional filters are 192, 256, 512, and 1024, respectively. Although it may not the best tailed implement, but it is capable to demonstrate our work. On the top of the convolutional blocks is 2 layers of full connection networks. The output layer of AttCNN contains 5 node that is same as the number of labels in these experiments.

By the way, we cite the result of original work, MIML (Zhou & Zhang, 2006) who published this dataset. MIML trained its model after a hand-crated pre-processing to extract feature vectors, which is different from the above 2 deep networks that learns directly on the raw image in an end-to-end manner. Nevertheless, comparing their result in detail helps us get a insight to discuss our model and feature work.

## 6 RESULT

### 6.1 SIMPLE EXAMPLE FOR FEATURE INPUT DATA

Table 2: The Results of Micro-F1 on the Feature Data Sets (Mean ± Standard Deviation)

| Dataset | PLST | MMOC | kNN | ML-kNN | LMMO-kNN | BiR |
|---------|------|------|-----|--------|----------|-----|
| scene | 0.5924±0.03 | 0.7297±0.03 | 0.7221±0.02 | 0.7387±0.03 | 0.7300±0.03 | 0.807789±0.025169 |
| corel5K | 0.0801±0.01 | - | 0.1006±0.01 | 0.0278±0.01 | 0.1834±0.02 | 0.343582±0.015272 |
| delicious | 0.1423±0.01 | - | 0.2338±0.01 | 0.1738±0.01 | 0.3046±0.01 | 0.249541±0.012315 |
| EUR-Lex (dc) | 0.2304±0.03 | - | 0.6988 ±0.01 | 0.6705±0.01 | 0.7388±0.01 | 0.684401±0.021701 |
| EUR-Lex (ed) | 0.1059±0.01 | - | 0.4664±0.01 | 0.3864±0.01 | 0.4706±0.02 | 0.53867±0.031739 |

Table 3: The Results of Example-F1 on the Feature Data Sets (Mean ± Standard Deviation)

| Dataset | PLST | MMOC | kNN | ML-kNN | LMMO-kNN | BiR |
|---------|------|------|-----|--------|----------|-----|
| scene | 0.4588±0.04 | 0.7025±0.03 | 0.6815±0.02 | 0.6874±0.03 | 0.7101±0.03 | 0.804263±0.025516 |
| corel5K | 0.0587±0.01 | - | 0.0773±0.01 | 0.0178±0.01 | 0.1517±0.01 | 0.195901±0.013047 |
| delicious | 0.1131±0.00 | - | 0.2094±0.01 | 0.1518±0.00 | 0.2665±0.00 | 0.253124±0.002821 |
| dEUR-Lex (dc) | 0.1530±0.02 | - | 0.6569±0.01 | 0.6038±0.01 | 0.7230±0.01 | 0.678230±0.010376 |
| EUR-Lex (ed) | 0.0725±0.01 | - | 0.4207±0.01 | 0.3385±0.00 | 0.4630±0.00 | 0.395483±0.012076 |

Results of the simple example of BiR are summarized in Table 2 and 3. For the feature input dataset, we report the Micro-F1 and example-based-F1 with the 10-fold validation on each dataset. The reason with using these measures was analyzed in Mao et al. (2012). Result for all other baseline method are taken from the paper Liu et al. (2018).

Here, the basic and simple example of our model merely acts as shallow and nonlinear projection from hand-crafted feature to metric space. Nevertheless, it addressed these multi-label metric learning problems and outperformed the baseline in most of the data sets.

### 6.2 BIDIRECTIONAL REPRESENTATION DEEP NEURAL NETWORKS

Table 4: The predicive accuracy On Multi-Label image data set scene 2000 with 5 labels

| labels\methods | MIML | AttCNN | BiRDNN |
|----------------|------|--------|--------|
| desert | 0.869±0.014 | 0.902142±0.013688 | 0.912812±0.018705 |
| mountains | 0.791±0.024 | 0.843571±0.024917 | 0.877187±0.018436 |
| sea | 0.729±0.026 | 0.713571±0.038502 | 0.792812±0.020572 |
| sunset | 0.864±0.033 | 0.943571±0.010271 | 0.951562±0.019469 |
| trees | 0.801±0.015 | 0.787142±0.018058 | 0.879687±0.026612 |
| overall accuracy | 0.811±0.022 | 0.827514±0.011797 | 0.882812±0.010703 |
| average precision | 0.783±0.020 | 0.850935±0.024998 | 0.862063±0.034688 |

We compare deep natural network implementation out model (BiRDNN) with AttCNN Hand et al. (2018) with identical convolutional network paper on raw scene image dataset. We also take the

result of MIML from the previous work Zhou & Zhang (2006) that learning the pre-processed feature on the same dataset instead of the raw data. Results are summarized in the Tables 4.

Because in the scene image data, there are much more negative samples than positive samples in all labels/tags, so AttCNN tends to mis-classify positive to negative, but tends not to mis-classify negative to positive. Therefore, false negative($fn$) samples is more than false positive($fp$) samples, so precision($tp/(tp + fp)$) is greater than accuracy $(tp + tn)/(tp + tn + fp + fn)$. MIML is likely not affected by the positive and negative distribution. BiRDNN uses the identical convolution component with AttCNN, but with the bidirectional representation it obtained better performance, and the metric learning predicts label via local information, the global skewed data distribution does not degenerate the result much.

## 7 DISCUSSION

### 7.1 MULTI-LABEL CLASSIFICATION MODEL

In the experiment demonstrate here, our simple example method for multi-label classification outperforms recent methods. Method based on output coding (Zhang & Schneider, 2012; Zhang & Zhou, 2007; Liu et al., 2018) are most likely limited to the capability of the feature mapping, or let say representation learning. The AttCNN (Hand et al., 2018) on the other hand is limited by the fact that it ignore the label dependency. Our proposed framework can overcome both limitations. In the framework of BiR, first we can implement the non-linear mapping easily just via 2 full-connection layers with an activation in the hidden layer. Moreover, the input and output representation share the same embedding when learning the metric space, thus the label dependency is exploited. For extensions, we can replace the full-connection networks by the more deeper convolutional to learn the representation for image data directly.

### 7.2 EXTENDIBILITY TO OTHER MULTI-OUT TASK

We have proposed and demonstrate a capable and extendable metric learning framework for multi-label classification(MLC). Our framework can extend to multi-target regression (MTR) where the binary label is replaced by real value as well as the multi-concept retrieval(MCR).

### 7.3 LIMITATION AND FUTURE WORKS

Here, we outline several limitations of our current work and how these might be overcome in future work.

(1) Testing time: After learning the discriminative and predictive metric space, we found the test stage is still not ideally fast enough using the on-the-shelf kNN tools. (2) Larger data set: Due the limited computational resource we can handle currently, we complete the experiment on the scene dataset with 2000 raw images that is not really large data set like the CelebA. Our future work will be conducted in larger and challenging data sets.

## 8 CONCLUSION

We have proposed a novel framework, BiR, for metric learning on multi-label data in including hand-crafted features and raw images. Our BiR frame uses an efficient bidirectional representation that incorporates label information when learning the discriminative and predictive metric space where input data are projected into. Experiments on a number of multi-label data sets suggest that the proposed BiR framework is capable of learning the representation for multi-label classification. In this setting, our model outperforms several recently proposed methods by a significant margin, while being extendable for more multi-output learning tasks and more complex applications.

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
