# OpenReview forum: "MULTI-LABEL METRIC LEARNING WITH BIDIRECTIONAL REPRESENTATION DEEP NEURAL NETWORKS"
_ICLR.cc/2020/Conference — Reject_

### Official Review · AnonReviewer3 · 2019-10-23
**Official Blind Review #3**

**Rating:** 1

**Review:**

The paper addresses the problem of multi-label prediction.  It proposes a method that uses a co-embedding of instances and labels into a joint embedding space in a way that related instances and labels fall close by and unrelated ones fall far away.  For this purpose, embeddings from input space and label space to a common space are learned from training data. At the prediction time, KNN to the embedding of the test instance in the co-embedding space is used to predict relevant labels.
Featurized (attributed)  labels are potentially considered, which can facilitate incorporating label dependence and generalization over unseen labels.

Main shortcomings:
- The novelty of the work is limited. Ideas introduced in this work are present and being investigated in literature for a while now, leading to remaining limited contribution for the paper.  Related work on joint embedding, co-embedding, label-embedding, and zero shot learning seem to be neglected totally. For example, the paper lacks awareness of, citation to and comparison with related work such as [1-5].
- Presentation of the paper can be highly improved. There are several grammatical and writing problems in the paper.
Formulation can also benefit from  improved presentation. See for example Eq (1).
- Technical arguments are not all well founded. For example, the scalability claim in the abstract  of the paper seems to refer to "prediction" time complexity being linear in the number of "training" examples, which is not actually fast.

In summary, based on the above reasons, I vote for the paper to be strongly rejected.

[1] Akata, Zeynep, et al. "Label-embedding for image classification." IEEE transactions on pattern analysis and machine intelligence 38.7 (2015): 1425-1438.
[2] Weston, Jason, Samy Bengio, and Nicolas Usunier. "Wsabie: Scaling up to large vocabulary image annotation." Twenty-Second International Joint Conference on Artificial Intelligence. 2011.
[3] Li, Xin, and Yuhong Guo. "Bi-directional representation learning for multi-label classification." Joint European conference on machine learning and knowledge discovery in databases. Springer, Berlin, Heidelberg, 2014.
[4] Mirzazadeh, Farzaneh, et al. "Scalable metric learning for co-embedding." Joint European Conference on Machine Learning and Knowledge Discovery in Databases. Springer, Cham, 2015.
[5] Yeh, Chih-Kuan, et al. "Learning deep latent space for multi-label classification." Thirty-First AAAI Conference on Artificial Intelligence. 2017.

**Experience Assessment:**

I have published in this field for several years.

**Review Assessment: Checking Correctness Of Derivations And Theory:**

I carefully checked the derivations and theory.

**Review Assessment: Checking Correctness Of Experiments:**

I assessed the sensibility of the experiments.

**Review Assessment: Thoroughness In Paper Reading:**

I read the paper thoroughly.

---

### Official Review · AnonReviewer1 · 2019-10-26
**Official Blind Review #1**

**Rating:** 3

**Review:**

This paper aims to solve multi-label problems via learning a share embedding space for instances and its label sets. Specifically, the author considers deep neural networks F(x) as an encoder for the instance (either raw input or features) and a shallow MLPs G(y) as an encoder for the label outputs. In the training stage, the instance embedding and its label embedding are forced to be close. An additional constraint is instances with different labels should be far from each other. After training, the inference can be done in the embedding space by looking up the labels of the query’s kNN instances.

Metric learning for multi-label problems is not new, many works have been proposed such as [1,2]. Using deep neural networks for multi-label problems is also not new, see [3,4]. Thus, the novelty of the proposed method is rather limited. The interesting part is the constraint of Eq(11), where kNN need to be updated whenever the instance encoder model F(X) is changing during the optimization, which is very expensive for large-scale application.

[1] Sparse Local Embeddings for Extreme Multi-label Classification, NIPS 2015.
[2] Learning Deep Latent Space for Multi-label Classification, AAAI 2017.
[3] Deep Learning for Extreme Multi-label Text Classification, SIGIR 2017.
[4] X-BERT: eXtreme Multi-label Text Classification with BERT, ArXiv 2019.


**Experience Assessment:**

I have published one or two papers in this area.

**Review Assessment: Checking Correctness Of Derivations And Theory:**

I assessed the sensibility of the derivations and theory.

**Review Assessment: Checking Correctness Of Experiments:**

I assessed the sensibility of the experiments.

**Review Assessment: Thoroughness In Paper Reading:**

I read the paper at least twice and used my best judgement in assessing the paper.

---

### Official Review · AnonReviewer2 · 2019-11-01
**Official Blind Review #2**

**Rating:** 1

**Review:**

This paper presents a metric learning approach for the multi-label classification problem. It basically maps the input features and the output labels into the same space and then uses k-NN to find the closest labels for each inputs. During training, it minimizes the squared Euclidean distance between the input embedding and label embedding. In the experiments, some image datasets and text datasets are used to compare with several multi-label learning algorithms.

The proposed method is clearly presented in the paper. However, I have several concerns.

1. The proposed method is straightforward and lacks of significant novelty.
2. The datasets are very small scale in terms of both sample size and label size.
3. The comparing methods are a little bit out-dated. I would also suggest to compare directly with the naive one-versus-all method using deep learning extraction models.
4. There are a few typos. For example, on Page 1, "because it wide application", "but cannon deal", "may lost key information"; on Page 2, "an bidirectional", "mapping of outputs to metric space are"; on Page 3, "a scalable models", etc.. Besides, Eq (1) should be written in a more formal way.

Overall, I vote for rejection and the main reason is the lack of novelty.

**Experience Assessment:**

I have published one or two papers in this area.

**Review Assessment: Checking Correctness Of Derivations And Theory:**

I assessed the sensibility of the derivations and theory.

**Review Assessment: Checking Correctness Of Experiments:**

I assessed the sensibility of the experiments.

**Review Assessment: Thoroughness In Paper Reading:**

I read the paper at least twice and used my best judgement in assessing the paper.

---

### Decision · Program_Chairs · 2019-12-19

**Decision:**

Reject

**Comment:**

All reviewers agreed that this submission is still premature to be accepted to ICLR2020.
We hope the review comments are useful for improving your paper for potential future submission.